# Experimental evidence of Willis coupling in a one-dimensional effective material element

Michael B. Muhlestein[1,2], Caleb F. Sieck[2,3], Preston S. Wilson[1,2] & Michael R. Haberman[1,2]

The primary objective of acoustic metamaterial research is to design subwavelength systems that behave as effective materials with novel acoustical properties. One such property couples the stress–strain and the momentum–velocity relations. This response is analogous to bianisotropy in electromagnetism, is absent from common materials, and is often referred to as Willis coupling after J.R., Willis, who first described it in the context of the dynamic response of heterogeneous elastic media. This work presents two principal results: first, experimental and theoretical demonstrations, illustrating that Willis properties are required to obtain physically meaningful effective material properties resulting solely from local behaviour of an asymmetric one-dimensional isolated element and, second, an experimental procedure to extract the effective material properties from a one-dimensional isolated element. The measured material properties are in very good agreement with theoretical predictions and thus provide improved understanding of the physical mechanisms leading to Willis coupling in acoustic metamaterials.

[1] Department of Mechanical Engineering, The University of Texas at Austin, 204 East Dean Keeton Street, Stop C2200, Austin, Texas 78705, USA. [2] Applied Research Laboratories, The University of Texas at Austin, Austin, Texas 78713, USA. [3] Department of Electrical Engineering, The University of Texas at Austin, Austin, Texas 78705, USA. Correspondence and requests for materials should be addressed to M.R.H. (email: haberman@utexas.edu).

The primary objective of acoustic metamaterial (AMM) research is to design subwavelength systems that behave as effective materials and display novel acoustical properties[1–3]. Examples of such novel properties include zero or negative dynamic mass density[4] and bulk modulus[5], chirality[6] and a more general material response that couples strain and momentum fields, as well as stress and velocity fields[7,8]. The latter response was initially described by Willis[9] and is therefore often referred to as Willis coupling. This behaviour may also be described as acoustic bianisotropy due to its mathematical similarities to bianisotropy in electromagnetism[8,10]. Although the initial theoretical description of Willis coupling was provided several decades ago, acoustic and elastic wave phenomena associated with the behaviour has received little attention until recently. One-dimensional constitutive relations with local Willis coupling may be written in the form[11,12]

$$\mu = \rho v + \tilde{\psi}\frac{\partial \varepsilon}{\partial t}, \qquad (1)$$

$$-p = \kappa \varepsilon + \psi \frac{\partial v}{\partial t}, \qquad (2)$$

where $\mu$ is the momentum density ($\mathrm{kg\,m^{-2}\,s^{-1}}$), $v$ is the particle velocity ($\mathrm{m\,s^{-1}}$), $p$ is the acoustic pressure (Pa), $\varepsilon$ is the volume strain (dimensionless) and the material properties are the mass density $\rho$ ($\mathrm{kg\,m^{-3}}$), bulk modulus $\kappa$ (Pa) and Willis coupling coefficients $\tilde{\psi}$ and $\psi$ ($\mathrm{Pa\,s^2\,m^{-1}}$). In general, these equations contain spatial and temporal convolutions[7–9].

Reciprocal Willis coupling results from two different physical phenomena: local coupling associated with microstructural asymmetry and nonlocal coupling associated with finite phase change across a unit cell and multiple scattering between spatially separated heterogeneities[12–18]. When constitutive relations for lossless media are written in the form of equations (1) and (2), the local and non-local contributions to Willis coupling manifest as the real and imaginary parts, respectively, of the coupling coefficients. Further, passivity requires that $\psi = \tilde{\psi}^*$ (refs 12,15,16). However, in lossy periodic media, both the real and imaginary parts of the coupling coefficient contain contributions from both local and nonlocal effects. Thus, for lossy systems it becomes difficult to distinguish between the two contributors to Willis coupling, but becomes easier when non-local effects are rendered negligible by considering an acoustically small metamaterial element that is isolated rather than part of an ensemble of mutually interacting elements. Under these conditions and with a properly designed experiment, coupling due to nonlocal effects may be neglected and the Willis coupling coefficients must be equal to each other, that is $\tilde{\psi} = \psi$, by reciprocity[12]. However, the extracted properties will only apply to the sample under study.

It is important to note that Willis constitutive relations do not uniquely describe the scattering from an effective material in the absence of a source distribution[10,15]. In other words, it is possible to represent a material response resulting from subwavelength heterogeneities and non-local effects with constitutive relations that are not of the form provided in equations (1) and (2). However, recent analogous work in electromagnetics demonstrated that neglecting bianisotropy in the effective material properties results in parameters that violate the principles of causality and passivity[12,19].

Previous theoretical work has shown that Willis coupling is absent in typical materials ($\psi = \tilde{\psi} = 0$) and, although various theoretical models have predicted the existence of Willis coupling, experimental observation of Willis coupling has only recently been reported in the work by Koo et al.[20]. Their work demonstrated a meta-atom consisting of concentric rectangular prisms with membranes of tunable thicknesses. Using coupled mode theory, a periodic system of meta-atoms was shown to predict the existence of acoustic bianisotropy. The structure proposed by Koo et al.[20] is innovative and was experimentally shown to display behavior associated with acoustic bianisotropy for an isolated unit cell and metasurface. Their results provide a valuable confirmation of theoretical predictions going back to the work of Willis[9]. However, Koo et al.[20] focused primarily on the utilization of acoustic bianisotropy for the control of acoustic waves. As a result, several aspects of Willis coupling that are of fundamental importance to understanding the physical phenomenon and its importance in AMM, such as the importance of including these coupling parameters, to predict and/or measure effective properties that are causal and passive, were not explored but are addressed here.

The study is structured as follows. First, a simple theoretical homogenization method is presented to provide insight into the physical origins of local coupling and to inform the design of an elementary effective material element that demonstrates local Willis coupling. Although the results of this model are not new, the derivation is unique and helps elucidate the physical mechanisms of Willis coupling. An extraction algorithm is then presented which makes use of reflection and transmission data. The algorithm is a generalization of the method published by Fokin et al.[21] and is similar to a procedure developed and used for electromagnetic materials[22]. An experimental apparatus used for a material property measurement is then described and a simple asymmetric effective material element is proposed. The need for the generalized extraction algorithm derived in this study when analysing the asymmetric effective material element is demonstrated by first using the conventional extraction method of Fokin et al.[21], which provides effective properties that depend on direction and violate passivity. Finally, the effective mass density, bulk modulus and Willis coupling coefficient estimates for the effective material element are shown to be in good agreement with a supplementary theoretical prediction and are consistent with physical restrictions based on passivity. The simplicity of the proposed effective material element and the theoretical derivations presented in this study clearly demonstrate the fundamental nature of Willis coupling and the necessity of including this parameter when describing the response of a broad class of AMM.

## Results

**Material properties from expansions by averages**. Material properties are macroscopic descriptors of overall microscopic behaviour and must therefore represent the microscale physics. In this work, microscale refers to material structure that is deeply subwavelength and macroscale refers to behaviour and properties associated with the overall response. For example, the mass density of a material is a representation of the relationship between the macroscopic average of the microscopic momentum density of a representative material element and its average velocity (another macroscopic measure of a microscopic quantity). As material properties represent relationships between averaged quantities (for example, momentum density and velocity), they are necessarily approximations of the microscopic physics. In this study, the material properties of a potentially heterogeneous element are determined using a continuum approximation and volume averaging techniques. This theoretical analysis provides a physical understanding of Willis materials and suggests a method by which an effective material element may be designed to exhibit significant Willis coupling.

The average momentum density and the average volume strain of a representative material element may be expressed

mathematically as

$$\langle\mu\rangle = \frac{1}{\Omega}\int_{\Omega}\mu(x')\mathrm{d}x' = \frac{1}{\Omega}\int_{\Omega}\rho(x')v(x')\mathrm{d}x', \qquad (3)$$

$$\langle\varepsilon\rangle = \frac{1}{\Omega}\int_{\Omega}\varepsilon(x')\mathrm{d}x' = -\frac{1}{\Omega}\int_{\Omega}\kappa^{-1}(x')p(x')\mathrm{d}x', \qquad (4)$$

where $\langle\cdot\rangle$ denotes a spatial average over the domain $\Omega$ of the element and $x'$ is the position. Although the integrands of equations (3) and (4) depend greatly on the details of the (possibly discontinuous) microstructure, they may be approximated as continuous and analytic if all length scales of interest are much larger than the largest length scale of the microstructure[23]. As shown in Supplementary Note 1, analytic functions may be expanded in terms of their volume averages and the volume averages of their derivatives. In addition, the volume average of an analytic function may be expanded in terms of the function itself and its derivatives evaluated at a given point. Using these two results, the expressions in equations (3) and (4) may be expanded and written as

$$\mu(x) = \langle\rho\rangle v(x) + \frac{\Delta x^2}{3}\left\langle\frac{\partial\rho}{\partial x}\right\rangle\frac{\partial}{\partial x}v(x) + O\big[(k\Delta x)^3\big], \qquad (5)$$

$$\varepsilon(x) = -\langle\kappa^{-1}\rangle p(x) - \frac{\Delta x^2}{3}\left\langle\frac{\partial(\kappa^{-1})}{\partial x}\right\rangle\frac{\partial}{\partial x}p(x) + O\big[(k\Delta x)^3\big], \qquad (6)$$

for a one-dimensional heterogeneous element, where $2\Delta x$ is the length of the material element, $x$ is the position of the center of the element and $k$ is the largest acoustic wavenumber of interest or spatial frequency within or around the element. It should be noted that in deriving these expressions the acoustic fields were restricted to being smoothly varying functions of position or $k\Delta x \ll 1$, which represents the limit applicable to metamaterials[1–3]. In addition, the constituent material properties were restricted to smooth functions of position. Although not necessary from a homogenization standpoint, this last restriction allows the results to be given in a form more amenable to physical interpretation. These equations may then be combined with the expression for the conservation of linear momentum, $-\partial p/\partial x = \partial\mu/\partial t$ (correct to $O[(k\Delta x)^2]$) and the definition of strain rate, $\partial\varepsilon/\partial t = \partial v/\partial x$, to yield constitutive relations for the heterogeneous medium in the form of the Willis equations:

$$\mu = \rho^{\mathrm{eff}}v + \tilde{\psi}^{\mathrm{eff}}\frac{\partial\varepsilon}{\partial t} + O\big[(k\Delta x)^3\big], \qquad (7)$$

$$-p = \kappa^{\mathrm{eff}}\varepsilon + \psi^{\mathrm{eff}}\frac{\partial v}{\partial t} + O\big[(k\Delta x)^3\big], \qquad (8)$$

where

$$\rho^{\mathrm{eff}} = \langle\rho\rangle, \qquad (9)$$

$$\kappa^{\mathrm{eff}} = \frac{1}{\langle\kappa^{-1}\rangle}, \qquad (10)$$

$$\tilde{\psi}^{\mathrm{eff}} = \frac{\Delta x^2}{3}\left\langle\frac{\partial\rho}{\partial x}\right\rangle = \psi^{\mathrm{eff}} + O\big[(k\Delta x)^2\big], \qquad (11)$$

$$\psi^{\mathrm{eff}} = -\frac{\Delta x^2}{3}\frac{\langle\rho\rangle}{\langle\kappa^{-1}\rangle}\left\langle\frac{\partial(\kappa^{-1})}{\partial x}\right\rangle, \qquad (12)$$

and the $x$ dependence of the field variables ($\mu$, $p$ and $\varepsilon$) has been suppressed for convenience. As suggested by the notation, one may identify the quantities in equations (9)–(12) as the effective

material properties based on a perfect knowledge of the microstructure of the heterogeneous medium. A full derivation of equations (7)–(12) is presented in Supplementary Note 2 where it is shown that $\psi = \tilde{\psi}$ for lossless ideal gases with linear disturbances. The predictions of the effective material properties given in equations (9)–(12) provide physical insight as to how a strongly coupled material element may be designed. It is worth noting that the expressions for $\psi^{\mathrm{eff}}$ and $\tilde{\psi}^{\mathrm{eff}}$ both depend on the average gradient of the microstructural material properties, reinforcing the idea that material asymmetry leads to local Willis coupling. Thus, the effective material element described below has been designed to have a large average gradient of mass density.

In practice, equations (9)–(12) do not provide a useful method to predict the effective material properties of a system. The reasons for this are, first, constituent material properties rarely vary smoothly in an effective material element and, second, metamaterial elements often make use of resonant inclusions with hidden degrees of freedom that can not be accounted for in the expansions used above. A more practical approach consists of assuming an appropriate set of constitutive relations and determining the effective material properties through the relations of the volume-averaged field quantities. If a purely mechanical system is known to possess microstructural asymmetry, then equations (1) and (2) may be used. By noting that the net force per unit volume acting on an effective material element is $\langle f\rangle = \partial\langle\mu\rangle/\partial t$, the volume averaged constitutive equations may be written as

$$\langle f\rangle = \rho^{\mathrm{eff}}\left\langle\frac{\partial v}{\partial t}\right\rangle + \tilde{\psi}^{\mathrm{eff}}\left\langle\frac{\partial^2\varepsilon}{\partial t^2}\right\rangle, \qquad (13)$$

$$-\langle p\rangle = \kappa^{\mathrm{eff}}\langle\varepsilon\rangle + \psi^{\mathrm{eff}}\left\langle\frac{\partial v}{\partial t}\right\rangle, \qquad (14)$$

and thus the material properties may be determined by

$$\rho^{\mathrm{eff}} = \left.\frac{\langle f\rangle}{\langle\partial v/\partial t\rangle}\right|_{\varepsilon=0}, \qquad (15)$$

$$\tilde{\psi}^{\mathrm{eff}} = \left.\frac{\langle f\rangle}{\langle\partial^2\varepsilon/\partial t^2\rangle}\right|_{v=0}, \qquad (16)$$

$$\kappa^{\mathrm{eff}} = \left.\frac{-\langle p\rangle}{\langle\varepsilon\rangle}\right|_{v=0} \text{ and} \qquad (17)$$

$$\psi^{\mathrm{eff}} = \left.\frac{-\langle p\rangle}{\langle\partial v/\partial t\rangle}\right|_{\varepsilon=0}. \qquad (18)$$

As the above relationships only require knowledge of the motion and pressure on the boundaries of a vanishingly small element, equations (15)–(18) are especially well suited for experimental methods, which generally do not have access to the internal fields of an element. The fact that effective material properties may be inferred only from knowledge of the boundaries of a small, isolated material element is the principle that underlies the experimental extraction method described below.

**Experimental extraction of Willis properties.** An experimental method of extracting effective material properties from effective material elements is required to verify the existence of Willis coupling and to validate the above theoretical predictions. Fokin *et al.*[21] demonstrated that the frequency-dependent complex-valued (lossy) mass density and bulk modulus may be extracted using the reflection and transmission coefficients measured from a one-dimensional effective material element. Their method assumes that standard material properties (mass

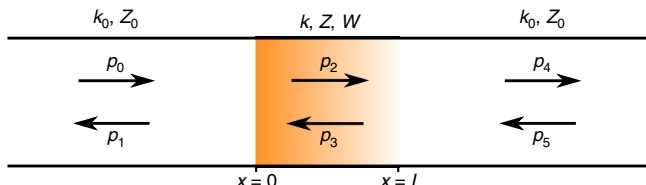

**Figure 1 | Schematic of a three-medium reflection–transmission problem.** The middle layer is assumed to be a Willis material with wavenumber $k$, characteristic impedance $Z$ and asymmetry coefficient $W$. The materials on the left and right are a uniform background medium with wavenumber $k_0$ and characteristic impedance $Z_0$. The pressures $p_0$, $p_2$ and $p_4$ represent right-propagating waves and the pressures $p_1$, $p_3$ and $p_5$ represent left-propagating waves.

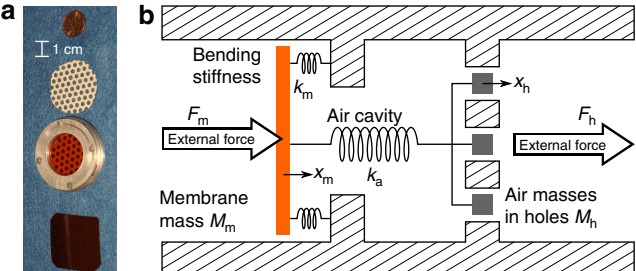

**Figure 2 | Photograph and mechanical schematic of the effective material element.** A photograph of the assembled element is shown second from the bottom of (**a**) along with membrane and perforated paper samples and a US 1-cent coin for scale. The membrane is 0.125 mm thick and the paper is 0.45 mm thick with 3.1 mm-diameter circular holes. The membrane and the paper are separated by a 5.9 mm-thick air cavity using an aluminum holder. The mechanical schematic shown in **b** describes the behaviour of the effective material element in the long-wavelength limit as a membrane with a mass $M_m$ and bending stiffness $k_m$, an air cavity with stiffness $k_a$ and air masses in the holes of a plate with mass $M_h$.

density and bulk modulus) fully describe the effective material element being measured. Although not obvious, a tacit assumption in using this approach is that the structure to be measured is symmetric about its centroid. A simple, if extreme, example that shows that systems without this symmetry may have different reflection coefficients is a finite rigid slab backed by a finite layer of vacuum. The transmission coefficient is equal to zero regardless of the side of incidence, but the phase of the reflection coefficient differs by 180°. In the symmetric case, the acoustic pressure reflection and transmission coefficients are the same regardless of the orientation of the effective material element with respect to the incident acoustic wave. However, it is shown below that if one does not take Willis coupling into account in the material extraction procedure, nonphysical lossy parameters will be measured for AMM with asymmetric microstructure.

To eliminate this error, one can generalize the approach of Fokin et al.[21] by assuming that the reflection coefficient depends on the incident wave propagation direction. This is achieved by measuring the reflection and transmission coefficients, $R$ and $T$, respectively, of a finite length one-dimensional effective material element in a background medium together with the reflection coefficient of the same effective material element in the reversed, or backwards, orientation, $R_B$ (the subscript B refers to the backward orientation). It is noteworthy that the transmission coefficient in both orientations is the same, $T_B = T$, when the effective material element is reciprocal.

A schematic of the situation under consideration is presented in Fig. 1 where the pressure field decomposed into forward and backward-propagating waves are represented by $p_i$; $k_0$ and $Z_0$ are the wavenumber and characteristic impedance of the background medium, respectively; and $k$, $Z$ and $W$ are the wavenumber, characteristic impedance and asymmetry coefficient, respectively, of the sample under evaluation. The effective material element parameters $k$, $Z$ and $W$ are related to angular frequency $\omega$, effective density $\rho$, bulk modulus $\kappa$ and Willis coupling coefficient $\psi$, by $k = \omega\sqrt{\rho/\kappa}$, $Z = \sqrt{\rho\kappa}$ and $W = \omega\psi/\sqrt{\rho\kappa}$. In Willis media, $Z$ and $W$ are components of the specific acoustic impedance, $Z_{sp}^{\pm} = Z(\pm 1 + iW)$ (assuming $e^{-i\omega t}$ time dependence). For a lossless Willis medium in the absence of non-local effects, $Z$ and $W$ are purely real. The fact that the specific acoustic impedance is always complex and changes phase angle with direction in media with asymmetric features is not commonly discussed or acknowledged in acoustics and necessitates the fundamental investigation and experimental validation this work provides. However, lossless periodic media with asymmetric unit cells have been shown to have complex Bloch impedances[24,25] and Kutsenko et al.[26] recently demonstrated that this same form relates to Willis coupling. It should be noted that specific acoustic impedances relate volume averaged quantities and do not depend

on the location of a unit cell boundary, whereas Bloch impedances relate the fields at the boundaries of a unit cell and do depend on the boundary locations[27].

Using the results of Supplementary Note 3 and the scattering coefficients $R = [p_1(0)/p_0(0)]_{p_5 = 0}$, $R_B = [p_4(0)/p_5(0)]_{p_0 = 0}$ and $T = [p_4(L)/p_0(0)]_{p_5 = 0} = [p_1(0)/p_5(L)]_{p_0 = 0}$ (using the notation of Fig. 1), the effective characteristic impedance, wavenumber and asymmetry coefficient may be written as

$$Z = \frac{Z_0 r}{(1 - R)(1 - R_B) - T^2}, \tag{19}$$

$$k = \frac{i \log(x)}{L} + \frac{2\pi m}{L}, \tag{20}$$

$$\text{and} \quad W = \pm \frac{R_B - R}{ir}, \tag{21}$$

respectively, where $r = \pm\sqrt{(1 - RR_B + T^2)^2 - 4T^2}$ and $x = (1 - RR_B + T^2 + r)/2T$. The sign of $W$ is positive in the quasi-static limit and at higher frequencies the sign is determined by requiring $W$ be a continuous function of frequency. The material properties are then found using the expressions $\kappa = Z\omega/k$, $\rho = Zk/\omega$ and $\psi = WZ/\omega$. As this analysis assumes reciprocity and that non-local coupling is negligible, the extracted values of $\psi$ are equivalent to the values of $\psi$. Although the reflection-transmission measurement does return boundary fields, for samples sufficiently small relative to a wavelength these fields may be used to approximate average fields and provide meaningful effective properties to describe the sample under study.

**Description of effective Willis material element.** To demonstrate Willis coupling, a plane wave tube-based experiment was used to determine the reflection and transmission coefficients of a one-dimensional asymmetric effective material element using both forward and backward orientations. Details of the experiment may be found in the Methods section. The 30 mm-diameter effective material element consisted of a 0.125 mm-thick membrane fabricated from DuPont Kapton FPC uniformly stretched across the sample holder, a 5.9 mm-thick air cavity and a 0.45 mm-thick perforated sheet of electrical insulating paper (Copaco) with 3.1 mm-diameter perforations yielding a surface void fraction of approximately 24%. The material properties of the Kapton FPC provided by the manufacturer are Young's modulus $E = 2.758$ GPa, Poisson's ratio

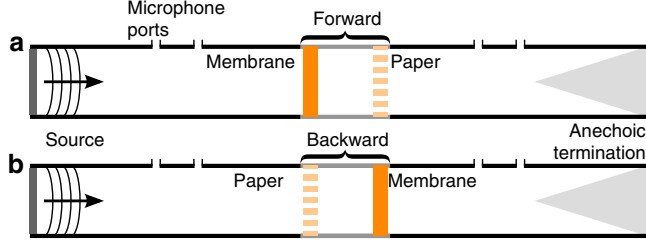

**Figure 3 | Schematic of the experimental apparatus with the effective material element in the forward and backward orientations.** The apparatus consists of a plane wave tube with a built-in source on one side, two microphone ports before and two microphone ports after the test section that contains the metamaterial element and an anechoic termination. The effective material element is placed in the test section in either the forward (membrane-air-paper) or backward (paper-air-membrane) orientation as indicated by **a,b**, respectively.

$v = 0.34$ and mass density $\rho = 1{,}420 \, \mathrm{kg \, m^{-3}}$. Figure 2 provides a photo of the effective material element and its long-wavelength lumped-element mechanical schematic, which is described below.

Owing to the relative simplicity of the effective metamaterial element, an analytical model may be developed to supplement the experimental results. It is well known that for sufficiently long wavelengths continuous dynamic elements may be well represented by lumped mechanical elements (such as springs and masses)[28–30], as demonstrated in Fig. 2 for the effective material element considered here. The values of the spring constants and masses used to model the effective material element as defined in Fig. 2 may be determined using the material properties and dimensions of the membrane, air cavity and air-filled holes in the perforated sheet. The work of Bongard et al.[31] provides a detailed model for the membrane mass $M_\mathrm{m} = 1.8830 \rho_\mathrm{m} h S$ and stiffness $k_\mathrm{m} = a^4/(192 S D)$, where $\rho_\mathrm{m}$ is the membrane density, $h$ is the membrane thickness, $a$ is the membrane radius, $S = \pi a^2$ is the surface area and $D = E h^3 / 12(1 - v^2)$ is the bending modulus[31]. For the membrane considered here, these become $M_\mathrm{m} = 0.23 \, \mathrm{g}$ and $k_\mathrm{m} = 27.1 \, \mathrm{kN \, m^{-1}}$ with $D = 5.08 \times 10^{-4} \, \mathrm{Pa \, m^3}$. Although the Bongard model of a membrane provides reasonable results (see, for example, Fig. 5 below), it does not account for pre-stress in the membrane or for losses in the system. The degree of agreement between model prediction and element behaviour can only be as good as the inputs to the model, and as the effects of the membrane tension and losses on the values of $M_\mathrm{m}$ and $k_\mathrm{m}$ are not known, an alternative approach to determining these parameters is to infer them from the measured data. The real and imaginary parts of $M_\mathrm{m}$ and $k_\mathrm{m}$ may be inferred from measurements of the effective mass density and then used for comparison with the other effective properties, as discussed more fully below. The movement of air in the holes in the perforated paper dominate the response of that component of the asymmetric element at low frequencies. The perforated paper may therefore be modeled as the mass of the air in the holes plus the added mass of the entrained fluid near them. The expression for the effective mass of this component of the effective material element is $M_\mathrm{h} = \rho_\mathrm{air} S (h + \Delta h) = 5.5 \times 10^{-3} \, \mathrm{g}$, where $\rho_\mathrm{air}$ is the mass density of air and $\Delta h = 16 a / 3 \pi$ is end correction to account for the entrained air[29]. Finally, in the long wavelength limit, the air cavity is dominated by its compressibility and may be modeled as a spring with stiffness $k_\mathrm{a} = S \rho_\mathrm{air} c_\mathrm{air}^2 = 16.8 \, \mathrm{kN \, m^{-1}}$, where $c_\mathrm{air}$ is the sound speed in air, which is assumed to be $346 \, \mathrm{m \, s^{-1}}$ (ref. 29).

The kinematics of the lumped-element model for the effective material element may be solved given an applied set of forces on the boundaries. Using this relationship between the kinematics

and the dynamics of the effective material element with equations (15)–(18) yields a theoretical prediction of the effective properties:

$$\rho^\mathrm{eff} = -\frac{k_\mathrm{m}}{\omega^2 V} + \frac{M_\mathrm{m} + M_\mathrm{h}}{V}, \qquad (22)$$

$$\kappa^\mathrm{eff} = \frac{L}{4S}(k_\mathrm{m} + 4 k_\mathrm{a}) - \frac{\omega^2 L}{4S}(M_\mathrm{m} + M_\mathrm{h}) \qquad (23)$$

$$\text{and} \quad \psi^\mathrm{eff} = \tilde{\psi}^\mathrm{eff} = \frac{k_\mathrm{m}}{\omega^2 2S} - \frac{M_\mathrm{m} - M_\mathrm{h}}{2S}. \qquad (24)$$

A full derivation of these properties is given in Supplementary Note 4.

**Measurement of Willis coupling.** Before analysing this asymmetric effective material element in terms of Willis material properties, it is motivational to first analyse the effective material element in terms of standard material properties that neglect Willis coupling. The experimental extraction technique of Fokin et al.[21] may be used to perform this analysis[21], which is equivalent to the extraction technique described above with $R_\mathrm{B}$ set equal to $R$. Using $R$ as the reflection coefficient leads to the forward estimates of the effective properties, and using $R_\mathrm{B}$ as the reflection coefficient leads to the backward estimates of the effective properties. The definitions of the forward and backward orientations for the effective material element considered here are given in Fig. 3.

As seen in Fig. 4, the two orientations of the effective material element yield noticeably different effective properties. The bulk modulus, in particular, shows remarkably different overall trends for both the real and imaginary parts for the different orientations. The bulk modulus in the backward orientation is nearly constant for the frequency range considered here, whereas in the forward orientation the bulk modulus decreases approximately linearly as a function of increasing frequency from 1,000 Hz to $\sim$1,400 Hz. For the forward orientation, the real part levels out around 150 kPa for higher frequencies, and the imaginary part continues to decrease. It is worth noting, in particular, that the imaginary part of the bulk modulus changes sign at $\sim$1,758 Hz. This is inconsistent with the requirements of reciprocity and passivity[12]. The imaginary part of the bulk modulus in the backward orientation also changes sign at this frequency, though the magnitudes are close enough to zero that this may be experimental error. This demonstration of unphysical material properties through the breaking of passivity is typical of published effective material properties where microstructural asymmetry is present but not accounted for[21,32,33]. The inconsistent and unphysical nature of the effective property estimates obtained by neglecting the effects of Willis coupling strongly suggests the need to account for Willis coupling when analyzing this effective material element.

The three experimentally inferred effective properties and the nondimensional asymmetry coefficient $W$ are presented in Fig. 5, along with the prediction associated with the simple model described above using the Bongard model to approximate the behaviour of the membrane (labelled 'Analytical membrane') and including the measured effective density to infer the membrane properties (labelled 'Inferred membrane'). The values of $M_\mathrm{m}$ and the real part of $k_\mathrm{m}$ for the 'Inferred membrane' approach are determined by fitting the predicted effective mass density in equation (22) to the real part of the effective density calculated from the experimental data at $f = 1 \, \mathrm{kHz}$ and at the density zero-crossing, which occurs at $\sim$1,713 Hz. The imaginary part of $k_\mathrm{m}$ is determined by fitting the model to the imaginary part of the effective density calculated from the experimental

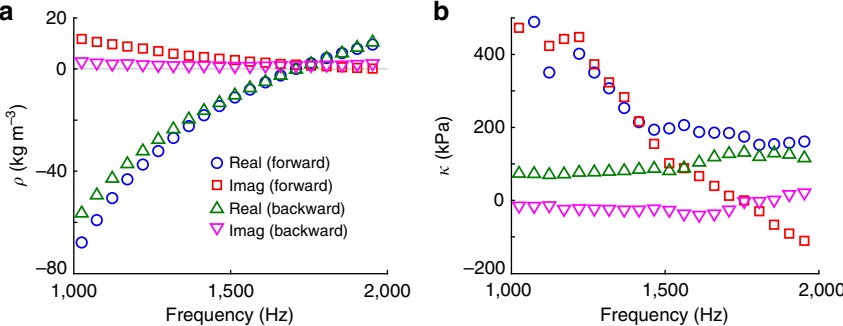

**Figure 4 | Extracted material properties of the effective material element when neglecting Willis coupling.** The effective density is shown in **a** and the bulk modulus is shown in **b**. Results are provided for both the forward and backward measurement configurations. The real and imaginary parts for the forward configuration are indicated by blue circles and red squares, respectively, whereas the green triangles and magenta inverted triangles indicate the real and imaginary parts obtained with the backwards orientation, respectively. The discrepancy in the extracted properties when testing in the different orientations illustrates the need to consider Willis coupling when subwavelength asymmetry is present.

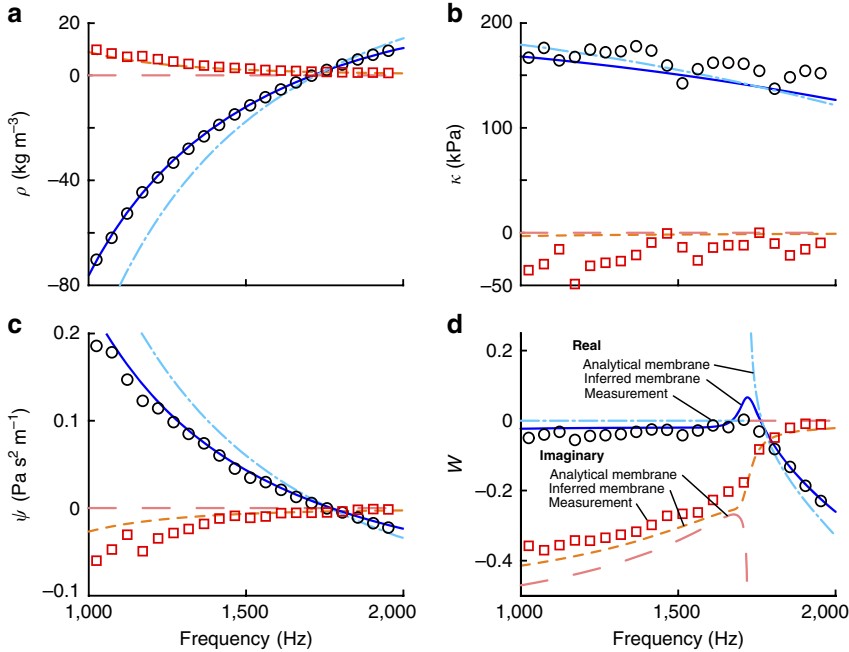

**Figure 5 | Predicted and measured effective properties of the effective material element.** The effective mass density $\rho$ is shown in **a**, the bulk modulus $\kappa$ in **b**, the Willis coupling coefficient $\psi$ ($= \tilde{\psi}$ by reciprocity) in **c** and the non-dimensional asymmetry coefficient $W$ in **d**. Two different predictions of the effective material element behaviour are shown based on the mechanical model developed in this work. One uses the analytical approach of Bongard[27] for the membrane properties using vendor-supplied material properties as model inputs and the dimensions used in the experimental setup. The real part of that model is indicated by dot-dashed lines and the imaginary part is represented by long-dashed lines. The second prediction is obtained by inferring the membrane parameters from the measured effective density values at 1 kHz and the frequency of zero effective density to get improved estimates of the effective behaviour using the same mechanical model. The real part of that prediction is indicated by solid lines and the imaginary part is represented by short dashed lines. The curve identification for each panel in this figure is the same.

data at 1 kHz and observing that the the imaginary part of the density nearly follows an $\omega^{-7/2}$ dependence. For these fits, the model parameters become $M_{\mathrm{m}} = 0.16\,\mathrm{g}$ and $k_{\mathrm{m}} = [19.3 - i1.5(f/1,000)^{-3/2}]\,\mathrm{kN\,m^{-1}}$, where the frequency $f$ is in Hz. The losses associated with the bending of the membrane dominate the losses of the heterogeneous element and may be treated as the only lossy component, as the other primary contributor to loss is the viscous loss associated with air flow through the perforated plate, which is estimated to contribute less than 1% of the losses of the membrane.

For all four quantities shown in Fig. 5, the 'Inferred membrane' predicted and measurement-extracted effective properties exhibit

very similar behavior. The predicted and extracted effective density values are very similar at all frequencies shown despite the model only being fit at two frequencies. The real part of the density is nearly $-80\,\mathrm{kg\,m^{-3}}$ near 1,000 Hz, increases smoothly to 0 around 1,713 Hz and continues on to positive values. The imaginary part of the effective density is about $9\,\mathrm{kg\,m^{-3}}$ at 1,000 Hz and decreases with frequency. The imaginary parts of the stiffness are negative and relatively close to zero (greater than $-50\,\mathrm{kPa}$) for the entire frequency range inspected, whereas the real parts are positive and around 150 kPa. The measurement data show relatively minor variations that do not appear in the model predictions. The oscillations occur roughly every 330 Hz,

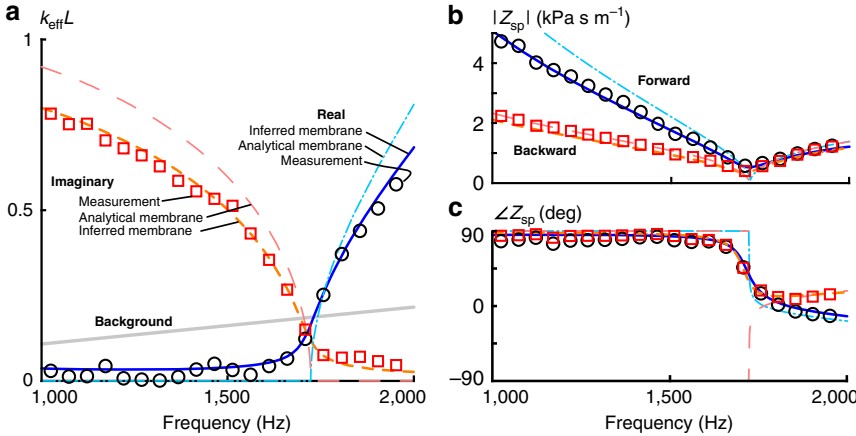

**Figure 6 | Measured and predicted effective wavenumber and specific acoustic impedance.** The effective wavenumber is shown in **a**, whereas the magnitude of the effective specific acoustic impedance and the associated phase in the forward and backward orientations is shown in **b,c**, respectively. The labelling of the line styles in **a** also hold for **b,c**. There appears to be a critical frequency, which is about 1,720 Hz. The effective wavenumber is almost entirely imaginary below this critical frequency and is almost entirely real above the critical frequency. The magnitude of the specific acoustic impedance is different for the two orientations below the critical frequency, but converge at a minimum value and increase nearly identically above the critical frequency. The phase of the specific acoustic impedance, on the other hand, is nearly 90° for both orientations well below the critical frequency, drops to about 0° at the critical frequency and splits symmetrically about zero based on orientation above the critical frequency. The splitting of the phase of the specific acoustic impedance based on orientation is indicative of the effect of Willis coupling on the specific acoustic impedance of the medium.

which corresponds to a wavelength in air of about 1.0 m or 1.3 impedance tube lengths and is therefore likely to be the result of a resonance due to the absence of an anechoic termination at the source end of the impedance tube. These variations also appear to a lesser extent in the extracted values of $\psi$ and $W$. The real parts of the Willis coupling coefficient are about 0.2 Pa s²m⁻¹ at 1,000 Hz and decay smoothly as a function of frequency and display a zero-crossing at ∼1,773 Hz. The imaginary parts are about $-0.5$ Pa s² m⁻¹ at 1,000 Hz and rise asymptotically towards 0. The real part of the asymmetry coefficient $W$ is nearly zero for frequencies less than about 1,720 Hz and then decreases nearly linearly to $-0.25$ by 2,000 Hz, whereas the imaginary part is about $-0.4$ at 1,000 Hz, increases slowly to about $-0.2$ by 1,720 Hz and then shows a rapid increase approaching 0 with increasing frequency.

## Discussion

The values of $W$ presented in Fig. 5 clearly demonstrate that Willis coupling must be accounted for when calculating the effective specific acoustic impedance of this effective metamaterial element. For example, at 2 kHz the specific acoustic impedance in the forward direction is predicted to be $Z = (1.2 - i0.26)$ kPa s m⁻¹ and in the backward direction is predicted to be $Z_B = (-1.1 - i0.33)$ kPa s m⁻¹, which has a phase ∼29° greater than $-Z$. Figure 6 presents a plot of the specific acoustic impedance in the forward and backward directions as a function of frequency. Throughout the frequency range presented, the magnitude or the phase of the specific acoustic impedance depends noticeably on the direction of propagation. Bradley[25] observed a similar directional dependence of Bloch impedances in a periodic waveguide with asymmetric unit cells, although he did not interpret the behaviour of the periodically arranged elements in the waveguide as an effective material and so did not attribute this phenomenon to Willis coupling. Notably, the splitting of the impedance in Fig. 6 above ∼1,720 Hz is very similar to the splitting shown in the lowest band depicted in Fig. 8 of ref. 25. The agreement of the two theoretical predictions with the experimentally obtained effective properties then strongly suggests that this system exhibits non-trivial Willis coupling.

Thus, despite the fact that Willis coupling may be termed a higher-order effect, it is clearly important when subwavelength asymmetry is present.

A few brief comments should be made on the interpretation of the effective behaviour of this simple effective material element as material properties. The first is in regards to element size and the validity of the effective medium approximation using boundary fields. One of the primary requirements of using homogenization methods to approximate a heterogeneous structure as an effective material is that the wavelength in the medium be much larger than a representative effective material element, or $kL \ll 1$. The predicted and measured effective wavenumber multiplied by the effective material element length are shown in Fig. 6, as well as for the background medium (air). All of the values of $kL$ and $k_0L$ are $<1$ throughout the frequency range of interest and are much $<1$ near the density zero-crossing around 1,730 Hz. Therefore, the effective wavelength is sufficiently long to consider the sample as an effective isolated material element with negligible non-local effects. For this case, volume averaged properties may be determined from the measured boundary fields. In addition, in this limit the constitutive relations may be written without convolutions.

The second important comment about this measurement and model is in regards to the inability to determine the response of an infinite array of effective material elements from the measurement of a single effective material element. As the measurement was made on a single effective metamaterial element, any multiple-scattering interactions that would take place in a material composed of large numbers of these elements are clearly omitted by both the model and measurement presented here. Thus, it is likely to be that placing two or more of these effective material elements in series would lead to a different set of effective properties, which is contrary to the concept of a material. Indeed, it has been shown that many unit cells are necessary for a periodic system to be correctly considered a material[34]. On the other hand, if several of these effective material elements were dispersed randomly within some matrix as inclusions and spaced far enough apart such that multiple scattering effects may be neglected, the macroscopic material properties of this metamaterial would be determined by an

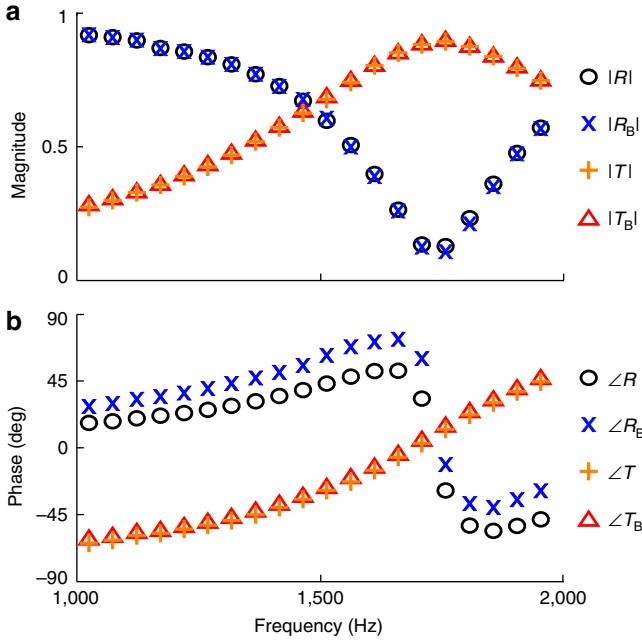

**Figure 7 | Scattering coefficients measured in the plane wave impedance tube.** The scattering coefficients for the forward orientation, $R$ and $T$, and for the backward orientation, $R_B$ and $T_B$, of the metamaterial sample. The magnitude and phase of the coefficients are shown in **a,b**, respectively. It is noteworthy that the sample is reciprocal and therefore the transmission coefficient $T$ does not change with sample orientation. In terms of the scattering coefficients, for this measurement the transmission coefficient $T = T_B$. However, the phase of the reflection coefficients, $R$ and $R_B$, are shown to be dependent on sample orientation.

average of the matrix properties and the effective inclusion material properties[35]. Then, in the sense of an isolated inclusion, the properties derived above may be treated as the effective material properties.

Finally, by comparing with effective properties extracted with conventional methods, the modified procedure accounting for Willis coupling was demonstrated to return physically meaningful effective properties. Traditional properties determined from a single reflection and transmission can only describe an effective material element in that exact measurement condition, and generally, one or more of the properties will not satisfy passivity and/or reciprocity restrictions. However, the modified procedure returned properties, which describe both orientations and satisfy physical restrictions for macroscopic parameters.

This study demonstrates experimentally and theoretically the need to account for local Willis coupling when determining physically meaningful material properties in one-dimensional asymmetric metamaterials. The results show that neglecting to account for Willis coupling leads to measured effective properties that depend on the orientation of the sample and are non-causal and non-passive. This paper also presents an experimental method to determine effective local Willis material properties of one-dimensional metamaterials. This experimental method has been demonstrated and compared with theoretical predictions with good agreement.

## Methods

**Plane-wave tube experiment.** A custom aluminum sample holder (shown in Fig. 2) was used to position two thin sheets of material around a 5.9 mm (± 0.2 mm) cavity of air between the main tube and the extension tube of a BSWA model SW477 impedance tube. Both the main tube and the extension tube have two microphone ports. The extension tube was terminated with a conical foam

insert to minimize end reflections. The absorption coefficient of this termination was measured to be above 0.9 for the frequencies of interest (1–2 kHz). The input signal was a swept sine waveform with start and stop frequencies of 1,000 and 2,000 Hz, and a total duration of 5.12 s, resulting in a spectral resolution of 0.195 Hz. Coherent time-domain averaging of 20 waveforms was used to improve the signal-to-noise ratio. Both excitation and acquisition were achieved using a DataPhysics Quattro and the associated software SignalCalc. The acoustic pressure signals were measured at each port of the impedance tube using a single PCB Model 130E21 6.3 mm (1/4 inch) pressure microphone. Transfer functions between each measured acoustic pressure signal and the drive signal were then calculated. A single sensor technique was used to eliminate the need for a sensor cross-calibration procedure[36]. For frequencies between 1 and 2 kHz, the measured coherence between the drive voltage signal and the microphone output signals was greater than 0.9985, implying a high signal-to-noise ratio and a linear system.

**Scattering coefficients.** The scattering coefficients were obtained using standard techniques[36]. Using the notation introduced in Fig. 1, we may define the ratios $A = p_1(0)/p_0(0)$, $B = p_4(L)/p_0(0)$ and $C = p_5(L)/p_0(0)$. In the limit that the terminations are anechoic, the ratios $A$ and $B$ are the scattering coefficients $R$ and $T$, respectively, and $C = 0$. The effective material element was then removed and replaced in the backward orientation (see Fig. 3), such that $\psi \to -\psi$, and the reflection and transmission coefficients were measured again. The scattering parameters were then determined using the relations

$$R = \frac{A - B_B C}{1 - CC_B}, \tag{25}$$

$$T_B = \frac{B_B - AC_B}{1 - CC_B}, \tag{26}$$

$$T = \frac{B - A_B C}{1 - CC_B}, \tag{27}$$

$$\text{and} \quad R_B = \frac{A_B - BC_B}{1 - CC_B}, \tag{28}$$

where the $B$ denotes the backward orientation. It is noteworthy that for reciprocal systems, $T_B = T$. This fact was verified for the measurements discussed in the Results section, as shown in Fig. 7, which presents the experimentally determined scattering parameters $R$, $R_B$, $T$ and $T_B$ as a function of frequency. In addition, for a lossless sample, $|R| = |R_B|$ and the phases of the two reflection coefficients always differ for an asymmetric sample.

**Data availability.** The data that support the findings of this study are available from the corresponding author upon request.

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

## Acknowledgements

This work was supported by ONR through MURI grant number N00014-13-1-0631.

## Author contributions

All authors contributed to developing the ideas, analysing the results and writing the manuscript. M.B.M. fabricated the test sample and executed the experiment. M.B.M. and C.F.S. developed the effective material model. M.R.H. and P.S.W. provided guidance on all aspects of the work.

## Additional information

**Competing interests:** The authors declare no competing financial interests.

