## [Peer Review File · Nature Communications]

Reviewers' comments:

Reviewer #1 (Remarks to the Author):

To cut to the chase, I have a very favorable opinion of this manuscript. It addresses a very timely topic, namely Willis coupling in acoustic metamaterials, in which I believe there is great current interest. It does so in a very clear, very readable, and very convincing fashion with a relatively simple structure (this is in contrast to the cited paper by Koo et al, which is certainly good work but from which it is hard to gain much insight and understanding of Willis coupling). I expect that this manuscript will be highly cited and highly influential no matter where it is published, and I strongly support publication in Nature Communications. I have only a few minor comments and suggestions that are listed below.

I find the $\psi/\tilde{\psi}$ notation of the two Willis coupling parameters confusing. This notation makes me expect that they are somehow related, like inverse complex conjugates of each other or something. But as near as I can tell they are not necessarily related. This may be standard notation, but could the authors make a small statement reminding the readers that they are independent parameters, and not necessarily connected despite the notation?

Personally, I find the standard retrieval approach that delivers different material properties for the two directions to be a very clear demonstration that the standard density-modulus parameterization is not sufficient even for simple structures. I wish this fact were mentioned earlier than the last sentence of the introduction. You might also consider putting figure 6 in front of figure 3, to establish early that the "standard" approach is unquestionably inadequate in this case.

Similarly, it would be helpful if the manuscript included some statement about how different R and R_B are. I realize there may not be space for a figure, but do they vary by 1%? 10%? 70%? And to convey the precision of the measurement it would be interesting to know how close T and T_B were in measurement (since they should be equal). Now that I think about it, perhaps these direct measurements could go in the Methods section? But even just a sentence or two with numbers would be helpful.

Reviewer #2 (Remarks to the Author):

This paper focuses on the Willis coupling in one-dimensional isolated asymmetric structure, analyzed in view of the homogenization theory.

Authors also experimentally validate their findings by using single local-resonant structure, to exclude ambiguity of source (nonlocal effect / loss) of the Willis coupling, with suggested retrieval method.

In detail, constructing upon the homogenization theory, authors derive Willis coupling coefficient, and discuss the physical origin / meaning of Willis coupling from the derived result.

Relation / differentiation of their findings to previous works are properly made. Excellent agreement to the experiment was also obtained, especially with the inclusion of the loss.

Nonetheless, while authors approach of 'homogenization theory for Willis coupling' certainly provides explicit and clear pictures for the acoustic bianisotropy, and is considered as a nice contribution in the study of AMM, however, this reviewer consider the scope of findings in this submission is too narrow, to be qualified for publication in Nat. Communications.

Most of all, the ultimate message of 'structural asymmetry' for Willis coupling have been claimed many

times before, using other platforms (membranes), approaches (CMT) or in other physical domains (electromagnetic, elastic). As well, effective parameter retrieval for bianisotropic media also has been treated before in other platforms, both in acoustics and electromagnetics domain (even if I agree on the fact that those were are not based on homogenization theory).

In fact, it could have been more educational, if authors have qualitatively assessed the competition of local vs. nonlocal effects; explicitly discussing the effect of source separation, as authors stressed in the introduction (e.g., by using the MM array of nonlocality, at least analytically). It would have been also interesting if authors have studied / proposed a structurally asymmetric structure of zero Willis coupling or, seemingly difficult, symmetric structure of non-zero Willis coupling, from their analytical results based intuition / reasoning.

To conclude, this reviewer believe that the current submission is a nice contribution - extending the homogenization theory to the acoustic bianisotropy, but also think that it would be hard to draw attentions of general readership, of Nature Communications. In below, I further discuss possible technical improvements which authors could consider, when resubmitting to other journals (such as Scientific Reports).

- It is not clear from the first reading of the paper, that the claim of this submission is in the 'homogenization theory' solution of the Willis coupling (providing better insights than other solutions), and its experimental validation. Make it clear.
- I believe that the section of 'Effects of Neglecting Willis Coupling', with Figs. 5&6 is better suited for 'Supplementary'.
- 'Perforated paper modeled as the mass of the air in the holes, the air cavity compressibility modeled as a spring' appears twice in the manuscript. Please check other redundancies in the manuscript.
- Authors' $\omega^{-7/2}$ dependent loss assumption (in inferred membrane model) need to be better justified / supported; by either numerical simulations or at least with reliable references.
- The oscillations in the spectrum (in Fig.3) is believed to be Fabry-Perot like oscillation, from the reflection at source-end input waveguide. Authors may consider placing absorber at the input (for better isolation of source).
- I believe use of labels (a,b,c,d) in Figs.3,4,6 will improve the clarity of explanation.
- There is no plot of ψ_{bar} in the experimental part. Meanwhile, it seems that there is apparent physical relation between ψ and ψ_{bar} . Could you provide the experimentally measured ψ_{bar} and also the relation between ψ and ψ_{bar} (if possible, e.g., in Supplementary section?).

Reviewer #3 (Remarks to the Author):

The authors present theoretical and experimental evidence of Willis coupling resulting solely from local behavior of a one-dimensional isolated element. I have some minor concerns and some major.

The description of the methods and techniques in the section "Exp. Extraction of Willis Properties" needs improvement. Little is described about Figure 1. What is k_0 , Z_0 etc. I can guess but it should be in there. What are p_i ? Do ρ and κ refer to the properties of the background medium or the effective properties? What is the significance of the normalized quantities W and $Z_{\text{sp}}^{\text{pm}}$? My second concern is that the relation between the section "Exp. Extraction..." and the succeeding Measurement section is not clear. Eqs. 19, 20, 21 give the effective properties in terms of the material and geometric properties of the membrane system directly. It appears that they do not require a transmission and reflection experiment of the kind described in the previous section. How exactly are Eqs. 8 and 9 used in the measurement section? Further concerns:

1. Why is it that the authors say that volume averages can be expressed in terms of boundary measurements? There is a relation between the two in the sense of the divergence theorem but necessarily in the direct form that the authors have used.

2. The authors should be careful about their interpretations here. Interpretations of effective properties from scattering data has resulted in a large number of less than accurate papers. See, for instance, Simovski's review on metamaterials. This must especially be true for interpreting Willis properties from scattering data. There are three issues here. First is that the Willis parameters are nonlocal which implies that they should ideally be expressed in a spatially convolution form (even if harmonic at a frequency) (Willis 1997 and several of his papers after that make it clear. See Srivastava 2015). The authors should comment as to why it is appropriate here to express the relationship as a simple multiplication. This might be okay but some comments are in order. Second issue is that the Willis properties are inherently non-unique and this emerges directly from the act of doing volume averaging in conjunction with the divergence theorem. Again, see the above papers for details. It would be useful for authors to comment on this in light of their experimental measurements. The third issue has to do with boundaries of a finite sample to which Willis properties are being ascribed. The issue was raised, although in a different context, by Willis himself in <https://arxiv.org/abs/1311.3875>, in his recent paper in JMPS (2015) and in his recent manuscript (<https://arxiv.org/abs/1610.09686>). I think it may be said that if Willis properties are being ascribed through a scattering measurement from a finite sample then those properties apply to that specific sample and (maybe even) that specific experimental configuration. To extend more universally one has to be careful and use transition layers (Drude) or jump conditions. This point does not negate the work in this paper but it behooves attention and comments from the authors.

Reviewer #1

- *I find the $\psi/\tilde{\psi}$ notation of the two Willis coupling parameters confusing. This notation makes me expect that they are somehow related, like inverse complex conjugates of each other or something. But as near as I can tell they are not necessarily related. This may be standard notation, but could the authors make a small statement reminding the readers that they are independent parameters, and not necessarily connected despite the notation?*

Response The authors thank the reviewer for their careful reading of the manuscript. The Willis coupling coefficients ψ and $\tilde{\psi}$ are indeed related (in fact, for a reciprocal system they are equal; see Ref. [15] of the manuscript). This point is implicitly mentioned, but the authors neglected to state it explicitly. Explicit mention of this relation is now included in the body of the article, and the supplemental materials includes a derivation that demonstrates that they are indeed equal in ideal gases.

- *Personally, I find the standard retrieval approach that delivers different material properties for the two directions to be a very clear demonstration that the standard density-modulus parameterization is not sufficient even for simple structures. I wish this fact were mentioned earlier than the last sentence of the introduction. You might also consider putting figure 6 in front of figure 3, to establish early that the “standard” approach is unquestionably inadequate in this case.*

Response The authors thank the reviewer for this insightful suggestion. The authors agree that the failure of the standard approach to predict consistent and physical estimates of the effective properties of the effective material element provides excellent motivation for the implementation of the more general extraction method. The section “Effects of Neglecting Willis Coupling” has been removed and its contents have been moved to just after the first paragraph of the section “Measurement of Willis Coupling”. Due to the structural nature of this modification, a number of additional modifications have been made to the structure of the “Measurement of Willis Coupling” section, while no content has been removed or added (aside from that explicitly described in this letter). The last paragraph of the introduction has also been rearranged accordingly.

- *Similarly, it would be helpful if the manuscript included some statement about how different R and R_B are. I realize there may not be space for a figure, but do they vary by 1%? 10%? 70%? And to convey the precision of the measurement it would be interesting to know how close T and T_B were in measurement (since they should be equal). Now that I think about it, perhaps these direct measurements could go in the Methods section? But even just a sentence or two with numbers would be helpful.*

Response The authors thank the reviewer for this insightful recommendation. A figure depicting the scattering coefficients and some text describing them have been included in the Methods section.

Reviewer #2

- *It is not clear from the first reading of the paper, that the claim of this submission is in the ‘homogenization theory’ solution of the Willis coupling (providing better insights than other solutions), and its experimental validation. Make it clear.*

Response The authors thank the reviewer for this helpful comment. The reordering of the latter sections described in response to the suggestions of Reviewer #1 resolve this concern. This reordering emphasizes the failure of the currently published effective

property extraction method, and clarifies the importance of this paper.

- *I believe that the section of ‘Effects of Neglecting Willis Coupling’, with Figs. 5&6 is better suited for ‘Supplementary’.*

Response The authors thank the reviewer for their desire to help clarify this paper. While the results of the section “Effects of Neglecting Willis Coupling” are not the primary result of this paper, the authors feel that these results are central to the motivation of this paper, which is to illustrate the necessity of including Willis coupling in the effective properties of acoustic metamaterials that have asymmetric microstructure in order to extract physically meaningful results. Furthermore, this comment is in opposition to the recommendation given by Reviewer #1, with whom the authors agree. Therefore, the authors respectfully disagree with this comment and have instead emphasized the content of the section “Effects of Neglecting Willis Coupling,” which we now feel more clearly communicates the contributions of this work.

- *‘Perforated paper modeled as the mass of the air in the holes, the air cavity compressibility modeled as a spring’ appears twice in the manuscript. Please check other redundancies in the manuscript.*

Response The authors thank the reviewer for their careful reading of the paper. The authors have carefully gone through the manuscript and have removed all redundancies that were found.

- *Authors’ $\omega^{-7/2}$ dependent loss assumption (in inferred membrane model) need to be better justified/supported; by either numerical simulations or at least with reliable references.*

Response The $\omega^{-7/2}$ dependence is not derived or predicted by models, but is the experimentally observed dependence. The wording of the original manuscript obscured this point. The phrase “...and assuming the imaginary part of the density follows an $\omega^{-7/2}$ dependence (as observed in the data)” has been changed to “...and observing that the the imaginary part of the density nearly follows an $\omega^{-7/2}$ dependence”. The authors acknowledge that this does not appear to be based on any specific physical behavior, and is likely due to multiple phenomena that contribute to loss in the system which may include membrane loss and leakage of acoustic pressure due to imperfections in the experimental apparatus. While this is indeed an ad hoc approach, we feel that it is sufficient for this study since the lossless model has good agreement with the experimental data, thus confirming the principle assertion of the work. However, the inclusion of loss based on experimental observations illustrates that a better characterization of the constituent materials and experimental apparatus is likely to lead to an improved agreement between model and measurement.

- *The oscillations in the spectrum (in Fig.3) is believed to be Fabry-Perot like oscillation, from the reflection at source-end input waveguide. Authors may consider placing absorber at the input (for better isolation of source).*

Response The authors thank the reviewer for their interest in this work. The authors had not previously considered putting additional absorption near the input, and in later iterations of this experiment may incorporate this idea. However, such refinements do not significantly add to the overall message of this paper, and so this observation is not included in the revised manuscript. Further, the revised manuscript now comments on the fact that reflections from the source end of the impedance tube are the mostly likely cause of the oscillations in the measured effective modulus.

- *I believe use of labels (a,b,c,d) in Figs.3,4,6 will improve the clarity of explanation.*

Response The authors thank the reviewer for this helpful suggestion for clarifying the figures of the paper. Labels have been added to the multi-axis figures showing data, as per request.

- *There is no plot of $\tilde{\psi}$ in the experimental part. Meanwhile, it seems that there is apparent physical relation between ψ and $\tilde{\psi}$. Could you provide the experimentally measured $\tilde{\psi}$ and also the relation between ψ and $\tilde{\psi}$ (if possible, e.g., in Supplementary section?).*

Response The authors thank the reviewer for their careful review of the paper. As discussed in response to the first comment from Reviewer #1, for any reciprocal material $\tilde{\psi} = \psi$, and this assumption is embedded in the homogenization scheme, so a graph of ψ is the same as a graph of $\tilde{\psi}$. These points have been made more clear in response to Reviewer #1's comment.

Reviewer #3

- *Little is described about Figure 1. What is k_0 , Z_0 etc. I can guess but it should be in there. What are p_i ? Do ρ and κ refer to the properties of the background medium or the effective properties? What is the significance of the normalized quantities W and Z_{sp}^{\pm} ?*

Response The authors thank the reviewer for their helpful suggestion. The variables presented in Figure 1 have been defined more clearly in the manuscript. Also, the variables ρ , κ , and ψ , which are the effective properties of the sample, density, bulk modulus, and Willis coefficient, respectively, are defined. W is the nondimensional asymmetry coefficient, which provides metric of the influence of Willis coupling on wave propagation. The specific acoustic impedance, Z_{sp}^{\pm} , is the ratio of pressure to particle velocity for a propagating wave. A medium with local Willis coupling will have a complex specific acoustic impedance, even when lossless, given by $Z_{sp}^{\pm} = Z(\pm 1 + iW)$. These properties have been clarified in section "Experimental Extraction of Willis Properties."

- *The relation between the section Exp. Extraction... and the succeeding Measurement section is not clear. Eqs. 19, 20, 21 give the effective properties in terms of the material and geometric properties of the membrane system directly. It appears that they do not require a transmission and reflection experiment of the kind described in the previous section. How exactly are Eqs. 8 and 9 used in the measurement section?*

Response The authors thank the reviewer for their comment to clarify the manuscript. The section "Experimental Extraction of Willis Properties" outlines the modified extraction procedure required to measure Willis coupling. Eqs. 8 and 9 (now Eq. 8 and 10) relate effective properties to the measured reflection, R and R_B , and transmission, T , coefficients. In order clarify the manuscript, the specific effective element that was measured is now introduced in the section titled "Description of Effective Willis Material Element," and a lumped element model is developed for comparison with the measured results. The effective properties from the lumped element model are provided in Eqs. 19-21 (now 18-20). The section "Measurement of Willis Coupling" now provides results and discussion comparing the lumped element model, experimentally extracted quantities using the method in the section "Experimental Extraction of Willis Properties," and a model fitted to the measured data.

- *Why is it that the authors say that volume averages can be expressed in terms of boundary measurements? There is a relation between the two in the sense of the divergence theorem*

but necessarily in the direct form that the authors have used.

Response The authors thank the reviewer for the suggested clarification. Because the truncation neglects $O[(k\Delta x)^2]$ and higher, the approximation of volume averages using boundary fields is valid for an acoustically small element with asymmetry, as demonstrated by the derivation in the supplementary materials. The text has been updated to emphasize this restriction.

- *The authors should be careful about their interpretations here. Interpretations of effective properties from scattering data has resulted in a large number of less than accurate papers. See, for instance, Simovskis review on metamaterials. This must especially be true for interpreting Willis properties from scattering data. There are three issues here. First is that the Willis parameters are nonlocal which implies that they should ideally be expressed in a spatially convolution form (even if harmonic at a frequency) (Willis 1997 and several of his papers after that make it clear. See Srivastava 2015). The authors should comment as to why it is appropriate here to express the relationship as a simple multiplication. This might be okay but some comments are in order. Second issue is that the Willis properties are inherently non-unique and this emerges directly from the act of doing volume averaging in conjunction with the divergence theorem. Again, see the above papers for details. It would be useful for authors to comment on this in light of their experimental measurements. The third issue has to do with boundaries of a finite sample to which Willis properties are being ascribed. The issue was raised, although in a different context, by Willis himself in <https://arxiv.org/abs/1311.3875>, in his recent paper in JMPS (2015) and in his recent manuscript (<https://arxiv.org/abs/1610.09686>). I think it may be said that if Willis properties are being ascribed through a scattering measurement from a finite sample then those properties apply to that specific sample and (maybe even) that specific experimental configuration. To extend more universally one has to be careful and use transition layers (Drude) or jump conditions. This point does not negate the work in this paper but it behooves attention and comments from the authors.*

Response The authors thank the reviewer for their insightful comments of Willis materials to clarify this work. The points that are raised by the reviewer are very important, non-trivial, and get at the fundamental difficulties of using Willis parameters to describe the overall dynamic response of heterogeneous media despite the very obvious need to include them as illustrated in this work. The primary point that is of importance here is that we have made every effort to simplify the element and test procedure to address the exact concerns that are raised by the reviewer. The motivation for doing so was two-fold. First, in order to get any experimental results that one can interpret “simply,” it was necessary to study a very simple system. Second, we believe that the only way to clearly communicate the need to include this particular type of coupling between effective momentum and pressure fields to a broad scientific audience was to find a very simple case that demonstrated the physical behavior of interest. We believe that the system we presented and justification for its study achieve those two objectives. The specific points raised by the reviewer are addressed in the following bullet points.

(i) On the nonlocality of Willis materials and representation with temporal and spatial convolutions, we agree with the reviewer and this is why the the experiment was limited to a single metamaterial element which was sufficiently small such that nonlocal effects, $O[(k\Delta x)^2]$ and higher, may be neglected. This has been emphasized throughout the revised manuscript. Additionally, as verified by comparing the measurement results, lumped element model, and field-averaging homogenization, ρ , κ , and ψ are well represented by leading order volume average properties of effective material element, and the necessity of including ψ when describing asymmetric elements was verified. Therefore, the constitutive

relations do not need convolutions. The dynamic nature of the effective material element studied are due to the resonant membrane, not higher order nonlocal effects.

(*ii*) As the reviewer noted, Willis material properties are inherently nonunique in the absence of sources. However, Alù (<https://doi.org/10.1103/PhysRevB.84.075153>) has shown that even though alternative material descriptions exist, only after accounting for bianisotropy (the electromagnetic equivalent of Willis coupling) will effective material properties satisfy passivity, causality, and reciprocity for a “homogeneous” medium. That not all macroscopic descriptions result in physically meaningful properties was demonstrated in the current work by using the traditional reflection and transmission measurement, and while these measurements provide macroscopic properties of the sample for a particular orientation, they violate passivity. Thus, the present work clearly demonstrates that the model accounting for Willis coupling will result in more physically meaningful properties.

(*iii*) On the boundaries of the effective material element and the extrapolation of the sample properties to larger arrangements, as suggested by the reviewer, the properties measured will apply to that specific sample. For 1D acoustic measurements, we may expect to extract “Bloch” properties for any number of complete unit cells and can be used to describe an infinite lattice or finite lattice with the same boundaries, as described in Simovski’s work; although, nonlocal effects will also need to be included to describe a larger lattice. If the unit cell is much smaller than wavelength then one may be able to consider these effective properties, as suggested by Simovski. This generally won’t be true for 2- and 3D lattices where transition layers will need to be taken into consideration. However, using the effective material element as an inclusion in a matrix, the macroscopic properties of the composite may be well approximated using the effective properties measured and the homogenization procedure of Muhlestein and Haberman (2016).

These issues have been addressed in a more straightforward manor in the introduction and in the concluding paragraphs of the Measurement section.

REVIEWERS' COMMENTS:

Reviewer #3 (Remarks to the Author):

I am satisfied by authors' response and recommend acceptance .